# Analyzing the comparative effectiveness of educational methods to foster children's dengue knowledge

**Maria Julia Hermida**[ID][1,2*], **Carolina Goizueta**[3], **Federico Giovannetti**[2,4], **Catalina Canosa**[3], **María Victoria Periago**[ID][2,3], **Carolina Lopez Ferloni**[3]

**1** Universidad Nacional de Hurlingham, Instituto de Educación, Villa Tesei, Provincia de Buenos Aires, Argentina, **2** Consejo Nacional de Investigaciones Científicas y Técnicas (CONICET), Argentina, **3** Fundación Mundo Sano, Argentina, **4** Unidad de Neurobiología Aplicada, Centro de Estudios Médicos e Investigaciones Clínicas "Norberto Quirno"- Consejo Nacional de Investigaciones Científicas y Técnicas (UNA, CEMIC-CONICET), Ciudad Autónoma de Buenos Aires, Argentina

* julia.hermida@gmail.com

## Abstract

Education about dengue disease is fundamental for its prevention. However, the most effective methods for delivering this knowledge in schools remain unclear. In this study, we compared two widely used educational interventions on their effectiveness in increasing dengue knowledge. A cluster-randomized controlled trial, with pre- and post-evaluations was applied. Three hundred ninety 10-year-old children from primary schools in a low-socioeconomic status area of Argentina that is endemic for dengue, were assigned to one of three groups: workshop, an abbreviated workshop, or direct instruction. Dengue knowledge was assessed with a questionnaire before, immediately after the intervention, and one month later. Teacher effect, concepts taught, duration of intervention, and baseline knowledge were controlled for. A Generalized Linear Mixed Model indicated that the three interventions were equally effective in improving dengue knowledge, both immediately after the intervention (children learned 3 dengue concepts) and one month later (children retained 2 dengue concepts). However, each intervention is more effective in teaching some concepts in particular. Our results, among the very few comparing educational interventions in dengue knowledge, suggest that the optimal approach depends more on contextual factors (e.g., specific concepts taught) than on the inherent superiority of either method.

## Introduction

Dengue is the most significant vector-borne disease affecting humans; globally, it is even more impactful than malaria in terms of morbidity and economic impact [1–3]. It is one of the fastest-growing infectious diseases worldwide, with an estimated 100–400 million new infections and approximately 20,000 deaths annually, predominantly

**Data availability statement:** The anonymized datasets, code files, materials, figures and tables are available from the OSF repository https://osf.io/hbdsc/?view_only=f-151c259e07146e6881454d5755ea18c.

**Funding:** This work was supported by Fundación Mundo Sano (no grant number available). The funders had no role in study design, data collection and analysis, decision to publish, or preparation of the manuscript.

**Competing interests:** The authors have declared that no competing interests exist.

affecting low socioeconomic-status populations [4]. In South America, the risk of dengue is expanding geographically, as well as the number of cases [5].

The most effective approach to dengue is prevention, which involves controlling the virus's vector, that in the case of Argentina, is the *Aedes aegypti* mosquito [6]. Preventive and control measures include, on the one hand, reducing potential mosquito breeding sites, and on the other, protecting against mosquito bites, for example, using repellents [7]. These practices require, at the individual level, the adoption of healthy habits to reduce the environmental burden of mosquito breeding sites, such as cleaning homes or emptying containers that could serve as breeding grounds. However, they also require community-wide cooperation: a single breeding site can trigger the spread of mosquitoes, for instance, from one garden to an entire neighborhood [6]. Education is key to effective behavioral change and community mobilization [8]. Thus, raising children's awareness about risks and preventive habits is essential for disease control [9].

Primary schools are the natural and ideal settings for disseminating this type of information [8,10,11]. Also, there is a greater need to provide education on dengue prevention in disadvantaged neighborhoods, as previous studies indicate that they often have less knowledge about the disease and exhibit fewer preventive practices [12]. However, there has been insufficient research on the most effective educational strategies for dengue in those communities [10], particularly in South America [13]. As a result, each country and school address this issue in various ways, often without measuring the effectiveness of their approaches.

Indeed, in countries like Argentina, where public health education is not part of the standard curriculum, regular teachers typically do not provide this knowledge. Instead, external experts are invited to schools to give talks to children on public health [11]. These talks typically employ a direct and structured instruction format, which focuses on providing students with the conceptual and procedural information necessary for specific learning [14]. While direct instruction boosts knowledge across various areas [15,16], including dengue [17], it has also faced criticism for potentially limiting creativity, autonomy, and critical thinking.

In response to these criticisms, alternative approaches have been developed and among them, one of the most widely used in health education is the workshop format [13]. It has been argued that the collective discussion and active participant involvement characterizing workshops would enhance the internalization of the content, including knowledge about dengue [13]. Unlike direct instruction, in a workshop setting, the teacher poses questions that guide students in the process of knowledge discovery [18]; students utilize their creativity and seek out new information to construct new understanding [19]. However, a critique of this approach is that it requires more time for students to fully assimilate the content, compared to other methods such as direct instruction.

There is an unresolved debate about which approach is more effective: direct instruction or active learning [14], with few studies addressing it. The case of dengue is particularly understudied. In fact, although numerous dengue interventions have been tested [20], most studies compare their impact against a passive control group that receives no intervention [21,22].

Among the few studies comparing the effectiveness of different interventions to increase dengue knowledge in primary schools, there are comparisons between board games versus theoretical classes [23], or versus providing teachers with theoretical material [24]. Another study compared two applications of the same board game in individual versus in-group [25]. The results from these previous studies suggest that the question of the effectiveness of the approaches has not yet been fully answered. Therefore, the current study aimed to contribute to filling this gap in the literature, comparing the effectiveness of approaches to increase knowledge about dengue in 10-year-old children in a dengue endemic area from Argentina.

## Materials and methods

### Design

A quasi-experimental design consisting of a factorial study (3x3) of repeated measures, including three evaluations [baseline (pre-intervention), T1 (immediate post-intervention), and T2 (one month after the intervention)] for three experimental groups (direct instruction, workshop, and abbreviated workshop) was conducted. Classrooms were randomly assigned to the groups. In each school, there was one classroom belonging to one study group and one belonging to another, but in no school were all three groups implemented since all had only two 4th-grade classrooms.

### Participants

The minimum sample size was calculated to ensure 80% statistical power using simulations based on effect sizes obtained in a previous study with similar characteristics and the same dependent variable [17]. The simulations were performed in R from RStudio (version 1.4.993, 2024). The initial calculation of the minimum sample size was 381, but the final sample was 390 cases.

The present project was approved by the Research Ethics Committee of the Province of Misiones (date of approval 23/12/2021). Parents or adults responsible gave written consent for their children's participation. Children gave their written assent for participation.

We included 10-year-old children to test the effectiveness of the devices because at this age they are literate, which allows both written knowledge assessments and workshops in which written resources can be used to work. In addition, the workshop material was pedagogically designed for 10-year-olds.

Children from 4th grade classrooms of all public schools in the city (11 schools) participated in the study. The schools were located on the outskirts of the city of Puerto Iguazú, Province of Misiones, Argentina, an endemic area for dengue fever. The children belonged to families with a low socioeconomic level: according to school records, 64% of the children's mothers were unemployed, housewives or supported by cash transfer programs. Data was collected between 01/08/22 to 03/08/23.

### Intervention

The three interventions were given by the same trainer, to control that there were no differences due to the teaching effect. To ensure fidelity to the intervention, research assistants were present in the three groups to verify that the 15 dengue concepts evaluated in the questionnaire were mentioned in the three interventions. The characteristics of the intervention are summarized in Table 1.

### Measures

Knowledge about dengue was assessed with a questionnaire used in a previous study [17], which was piloted again before the present study. The questionnaire consists of 15 items (S1 Table) with two response options: true or false. The three versions of the questionnaire have the same items but paraphrased and in random order. The number of correct true and false responses was similar (8 true, 7 false) in all versions. Children were randomly given one of the three versions

**Table 1. Summary of the characteristics of each intervention (direct instruction, abbreviated workshop, workshop) used to provide children with information on dengue prevention in Puerto Iguazú, Misiones, Argentina.**

| | Direct instruction | Abbreviated workshop | Workshop |
|---|---|---|---|
| Educational approach | direct instruction | workshop (time limited) | workshop |
| Planned duration (in minutes) | 40 | 60 | up to the expert consideration |
| Activities and materials | talk (showing images) | puzzle | puzzle |
| | | word-guessing game | word-guessing game |
| | | storytelling (attentional game) | storytelling (attentional game) |
| | | recognizing prevention situations game | board game |
| | eggs and larva visualization | eggs and larvae visualization | eggs and larvae visualization |
| Video with synthesis of concepts | yes | yes | yes |
| Dengue brochure | yes | yes | yes |

of the True-False test, a pen, and 10 minutes to complete the test. For each child, the version given at each time was different, in order to avoid the test-retest effect. The final score ranged from 0 to 15 and was the sum of correct answers. Subsequently, those protocols with scores below 5 (i.e., those with correct answers in only 30% of the questionnaire) were excluded from the analysis because they were considered unreliable. No behavioral change measure was included.

## Analysis

To assess the similarity of the different questionnaire versions in terms of scores and resolution times, we used a Kruskal-Wallis H test. The same test was implemented to compare the duration of the intervention activities and the number of participants included in them.

To test the hypothesis that the workshop group would increase dengue knowledge significantly more than the other study groups, and that they would maintain these changes over time, test scores were compared at baseline, T1, and T2, by study group, using a Generalized Linear Mixed Model (GLMM). In such a model, the dependent variable was the score at T2. Our factor of interest (group) and the baseline measurements were entered as fixed effects factors in the model. Case (i.e., participant id) was included as a random effects factor for the intercept. Model assumptions were checked by visually inspecting heteroscedasticity and normality of residuals, as well as normality of random effects (S1 Fig).

In addition, a second analysis to verify the results was performed by running another GLMM for each question in the questionnaire. This allowed determining whether any of the groups (direct instruction, workshop, or abbreviated workshop) was better for any particular question(s). In this case, each question in the questionnaire was the dependent variable and the independent variables were time, group, and time and group combined. The F-statistic was used for all GLMMs.

Moreover, given that GLMMs allow us to control unbalanced group sizes, there were children in T1 and T2 who did not take part in baseline, and vice versa, children in baseline that did not take part in T1 or T2. The percentage of children by group, at each time, that were also evaluated in other two timepoints was calculated.

## Results

In total, 390 children participated in the study. Table 2 shows the number of children per group. Although there were more participants in the workshop group, there was more than 80% overlap between the children who participated in each evaluation, with 17.34% of children in T1 and T2 not included at baseline (S2 Table).

There were no statistically significant differences in scores between the versions of the questionnaire (p = 0.28) nor in the resolution times taken by each version (p = 0.22). That is, the versions of the questionnaire used are valid for comparisons.

**Table 2. Descriptive statistics for each study group (direct instruction, abbreviated workshop, workshop) and assessment time (Baseline, T1, T2), Puerto Iguazú, Misiones, Argentina.**

| Group | Baseline | | | T1 | | | T2 | | |
|---|---|---|---|---|---|---|---|---|---|
| | n | Mean | SD | n | Mean | SD | n | Mean | SD |
| Direct instruction | 120 | 9.267 | 2.028 | 124 | 11.952 | 2.237 | 101 | 11.000 | 2.236 |
| Abb. workshop | 112 | 9.196 | 1.888 | 116 | 12.224 | 2.191 | 98 | 10.806 | 2.345 |
| Workshop | 143 | 9.007 | 1.915 | 146 | 12.253 | 2.283 | 124 | 11.355 | 2.457 |

n: Number; Mean: Score mean; SD: Score standard deviation; Abb: abbreviated

Note: Sample sizes variations across times are mostly because baseline and T1 data were collected during the same day, while T2 data was collected one month later.

The duration of the direct instruction (M = 37.64 min, SD = 5.42) was significantly shorter (p < 0.001) than the abbreviated workshop (M = 52.61 min, SD = 4.21) and workshop (M = 55.11, SD = 8.01), as predicted. However, the abbreviated workshop and workshop did not differ from each other in terms of time. At baseline, the three study groups showed similar knowledge about dengue (H = 1.816, p = 0.403). In addition, the number of participants per activity did not differ significantly between groups (H = 1.874; p = 0.392).

The GLMM to test the main hypothesis showed that the scores had statistically significant changes between the different evaluation times (Table 3) and that there were no differences between groups (p > 0.20). Specifically, participants significantly improved their knowledge of dengue between baseline and T1 in all groups (p < 0.001). Likewise, after one month, children reduce their knowledge but do so in the same way in all three groups (p < 0.001). In any case, this reduction in knowledge does not reach the baseline level (Fig 1): this indicates that the three interventions, i.e., direct instruction, abbreviated workshop, and workshop, are equally effective in improving knowledge about dengue.

To check the GLMM results, delta comparisons were performed and yielded similar results, they did not differ between groups for either T1 (F = 1.349; p = 0.26) or T2 (F = 2.893; p = 0.06).

In addition, GLMM was applied to determine whether either approach was better for certain questions. The results indicated that in 11/15 questions there were no significant differences. Significant differences were found in four questions. First, on the question *"Do mosquitoes bite more on the face and back than on the legs and arms?"* (Fig 2A), the Workshop group showed a statistically significant decrease in performance between T1 and T2 (β = −0.63, p = .03). Second, in the item *"Besides mosquitoes, can people also spread dengue fever?"* (Fig 2B), at T1 the direct instruction group showed statistically significant differences concerning the workshop group (β = −1.70, p < .001), the abbreviated workshop group

**Table 3. Generalized mixed model results and model fitness values.**

| Coefficient | Estimate | SE | df | t value | Pr(>|t|) | 2.5% | 97.5% |
|---|---|---|---|---|---|---|---|
| Intercept | 9.12 | 0.48 | 590.7 | 19.025 | < 0.001 | 8.19 | 10.06 |
| Time | −1.14 | 0.54 | 362.11 | −2.105 | 0.036 | −2.20 | −0.08 |
| Abb. workshop | 0.28 | 0.28 | 582.89 | 0.998 | NS | −0.26 | 0.83 |
| Workshop | 0.34 | 0.26 | 581.76 | 1.273 | NS | −0.18 | 0.86 |
| Baseline | 0.31 | 0.04 | 593.05 | 6.392 | < 0.001 | 0.218 | 0.41 |
| Baseline: Time | 0.01 | 0.05 | 363.17 | 0.263 | NS | −0.09 | 0.12 |
| Time: Abb. workshop | −0.34 | 0.30 | 343.97 | −1.12 | NS | −0.95 | 0.25 |
| Time: Workshop | 0.20 | 0.29 | 343.19 | 0.70 | NS | −0.36 | 0.77 |

*Note.* In this model, direct instruction group is the reference group. $R^2$ = .76; AIC = 3065.7; BIC = 3111.3. df = degrees of freedom; SE = standard error. NS indicates p value > 0.05, i.e., non-significant associations.

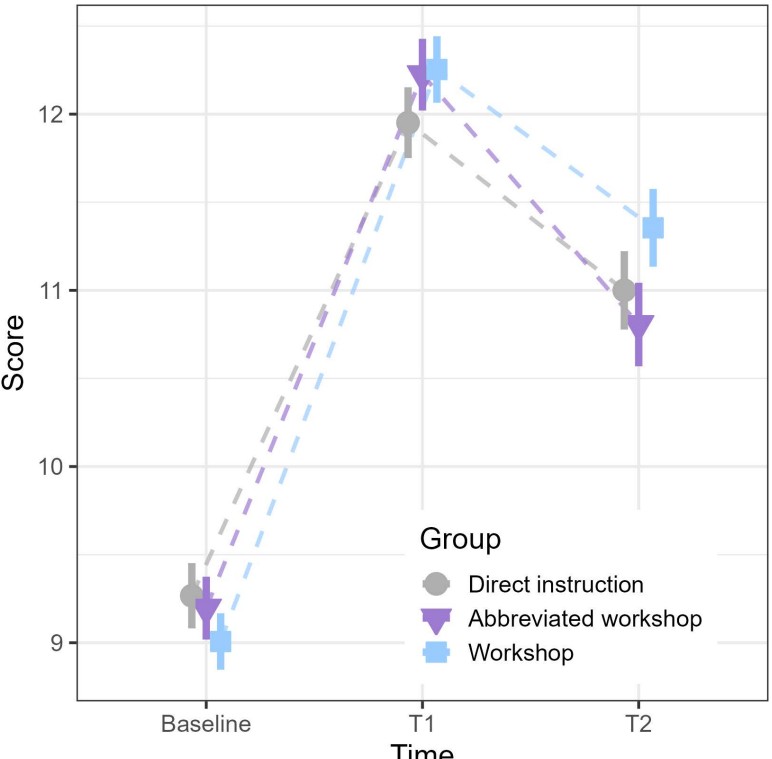

**Fig 1. Knowledge about dengue for each study group and time point (median, standard error and adjustment line), Puerto Iguazú, Misiones, Argentina.**

also showed differences but marginally significant ($\beta=-0.87$, p=.05). At T2, the direct instruction group maintained the differences ($\beta=-1.12$, p=.01). In addition, the direct instruction ($\beta=-1.09$, p=.01) and abbreviated workshop ($\beta=-0.83$, p=.02) groups decreased in the proportion of correct responses at T2 relative to T1. Third, in the item *"Before flying, is the mosquito a larva that lives in water?"* (Fig 2C), the direct instruction group and the workshop group showed a higher proportion of correct answers at T2 than at T1 ($\beta \geq 2.40$, $p \leq 0.01$), while the abbreviated workshop group showed a decrease in those values ($\beta \leq -3.40$, p<0.001). The direct instruction and workshop groups showed lower performances than the abbreviated workshop group at T1 ($\beta \leq -2.04$, $p \leq 0.006$) and higher at T2 ($\beta \geq 3.78$, p<0.01). Finally, in the item *"Is the aedes aegypti black with white spots and stripes on its body and legs?"* (Fig 2D), the direct instruction and workshop groups showed lower values in T2 than in T1 ($\beta \leq -0.17$, p<0.001). The workshop group had higher values than the direct instruction group at both T1 ($\beta=0.03$, p<0.001) and T2 ($\beta=1.34$, p<0.001).

## Discussion

Education is one of the central pillars of dengue prevention [12]. While there are several educational interventions that have demonstrated effectiveness compared to passive control groups [21,22], very few studies have compared interventions against active control groups. In other words, little research evaluates the efficacy of two educational interventions in improving knowledge about dengue.

The study herein aimed to address this gap in the literature, by comparing the efficacy of two instructional approaches that are often contrasted in the field of education [14]: direct instruction and workshops. Our findings indicated that the direct instruction, the workshop, and the abbreviated workshop groups had the same positive impact on dengue-related

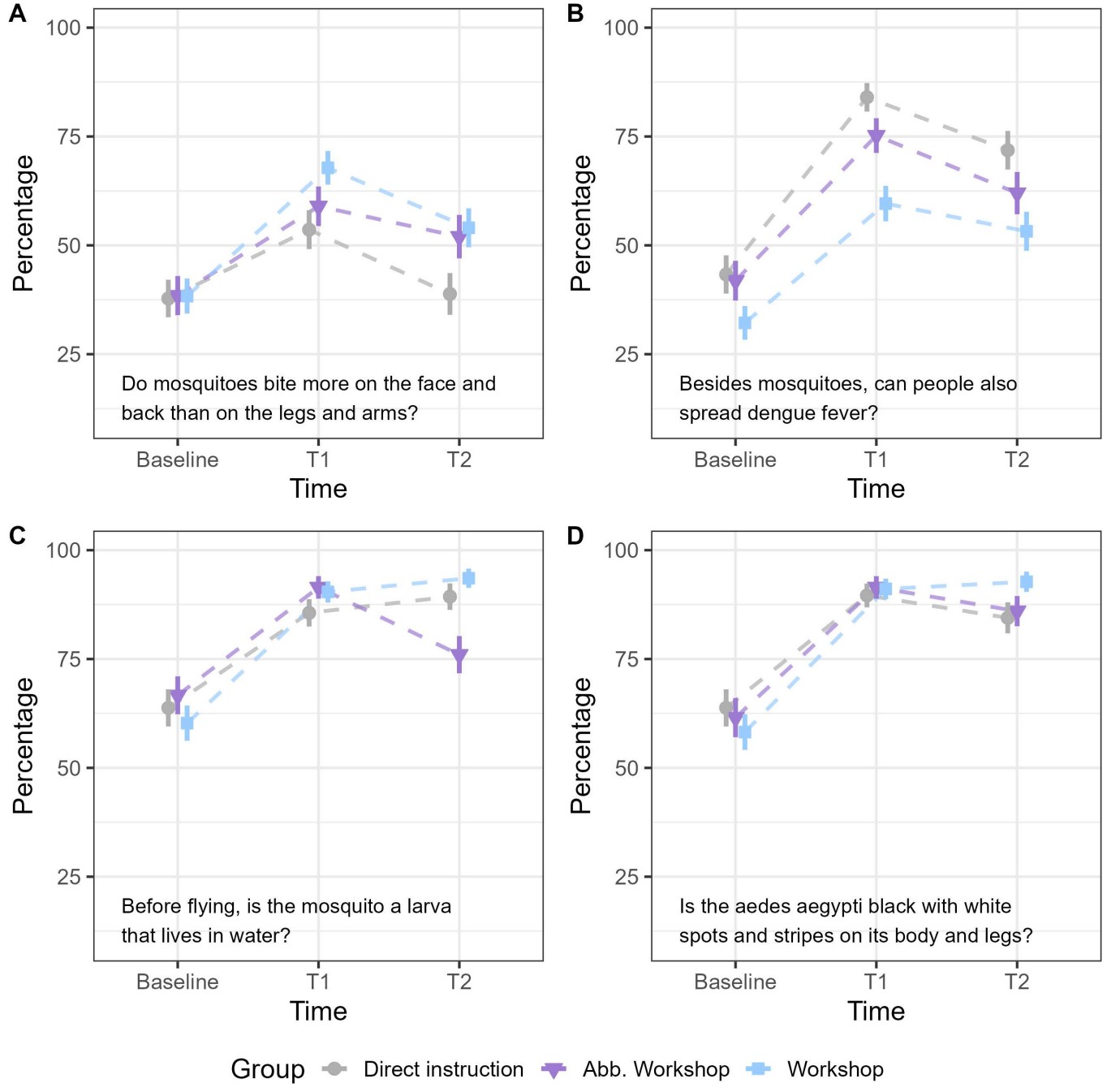

**Fig 2. Percentage of correct responses for questions with statistically significant differences between or within groups (median, error, and adjustment line).**

knowledge among 10-year-old children from a low-socioeconomic status and dengue-endemic region. In all three cases, the children improved their average knowledge from 9.15 (SD = 1.94) concepts to an average of 12.14 (SD = 2.24) concepts, increasing their dengue knowledge in three concepts from baseline to T1. After one month, the children in all three groups retained an average of 11.05 (SD = 2.35) concepts about dengue, meaning that in T3 they knew two

more concepts than in baseline. In other words, all three educational approaches had a similar effect: they significantly increased dengue-related knowledge compared to baseline levels.

Contrary to expectations, the workshop approach did not prove significantly better than the direct instruction in terms of dengue-related knowledge; rather, both methods yielded equivalent results. While this type of study does not provide precise explanations for the observed outcomes, in the specific case of dengue, previous research has yielded similar results, indicating that group and individual approaches are generally equally effective [25]. Importantly, group approaches such as workshops may enhance other variables (e.g., motivation), which were not examined in our study. Future research should be conducted to explore these potential factors further. The data from the present study only allow us to conclude that the three approaches are practically equally effective and that this outcome is not influenced by the facilitator conducting the interventions, the concepts covered in each intervention, or the group's prior knowledge, as these factors were controlled for in our study.

Moreover, if we assess the effectiveness of the different educational strategies for each specific item, we find significant differences only for four of them. First, regarding the areas of the body where mosquitoes bite most frequently (i.e., "*Do mosquitoes bite more on the face and back than on the legs and arms?*"), the workshop was more effective than the direct instruction, although these differences disappeared after one month. This result is partially opposite to the one found by Lennon and Combs [23], who found the direct instruction was better than a board game. One possible explanation for this could be that the workshop dedicated more time to reflecting on this point, presenting it as a real-life scenario where children were asked whether it was true or false that mosquitoes tended to bite in specific areas. By presenting a concrete situation related to this point, the workshop may have encouraged deeper reflection and stronger associations with real-life examples compared to the other groups.

Conversely, this time in line with Lennon and Combs [23] result, the direct instruction was more effective than the workshop in reinforcing the idea that dengue is not transmitted from person to person (i.e., "*Besides mosquitoes, can people also spread dengue fever?*"), both immediately after the intervention and one month later. A possible explanation for this result lies in the fact that the direct instruction included a specific image dedicated to illustrating this point, whereas in the workshops, this was merely one question among many others.

On the other hand, the abbreviated workshop was more effective than the other two groups in teaching the concept that larvae develop into mosquitoes (i.e., "*Before flying, is the mosquito a larva that lives in water?*") immediately after the intervention. Complementarily, the direct instruction and the workshop were more effective than the abbreviated workshop group one month later. Future studies should aim to explain the specific dynamics of this particular item.

Finally, the workshop group was more effective than the direct instruction group in teaching the children about the appearance of the *Aedes aegypti* mosquito (i.e., "*Is the Aedes aegypti black with white spots and stripes on its body and legs?*"), both immediately after the intervention and one month later. This result is also opposite to the one found by Lennon and Combs [23], and one potential explanation is that in the materials used during the workshop, there was a puzzle featuring an image of a mosquito, which took approximately ten minutes to complete. Given that the puzzle depicted an *Aedes aegypti*, it is reasonable to expect that the children's attention was focused on the physical characteristics of the mosquito, thereby reinforcing this information more effectively.

In summary, while care was taken to ensure that all concepts and the video summarizing them were presented in all three modalities, it is possible that, due to their inherent dynamics, each modality emphasized certain concepts more than others, allocating more time or providing more detailed explanations for specific points, as reported in previous studies [23,25]. This may have resulted in each approach having its own particular advantage. However, the minimal number of differences found and the fact that these differences favored different groups suggest that, overall, both direct instruction and the workshop modality are equally effective in increasing dengue-related knowledge among 10-year-old children. Future studies could explore whether combining both approaches might yield even greater effectiveness than each implemented separately. In addition, the generalizability of our results to other age cohorts requires further investigation.

 

As knowledge retention capacity is critically influenced by age, particularly due to the rapid neurodevelopment occurring during primary school [26], future studies should explicitly test whether our findings can be replicated across childhood. Last but not least, knowledge on dengue prevention is necessary, but not enough to produce behavioral change, which is ultimately what is required to prevent the spread of dengue. The lack of behavioral change measures is a limitation of our study that should be further explored.

These results hold theoretical significance for the field of dengue education, primarily because they contribute to the limited body of studies comparing the effectiveness of more than one educational strategy [10,13]. Specifically, our findings suggest that the choice of method may depend more on contextual factors, such as the specific concepts being taught, rather than on a presumed superiority of one approach over the other.

Also, our findings inform the planning and implementation of dengue health education, not only in Argentina, but also in other countries with areas endemic for dengue. Current dengue education efforts in Argentina employ a methodology that is selected at the discretion of the educator and usually not grounded in empirical evidence. This study informs educators, whether from NGOs, as in the present case, the public sector, or both, about the comparative benefits of utilizing one pedagogical approach over another. It thereby empowers them to select the methodology they deem most appropriate for a specific context while ensuring this decision is evidence-based. Remarkably, most educators in dengue prevention have traditionally used a direct instruction method to provide information on transmission and breeding sites of the vector to children in elementary schools. More recently, workshop methodologies were incorporated. By demonstrating that both direct instruction and workshops are equally effective, and similar in magnitude (approximately two additional concepts retained after one month), our results provide policymakers and educators with flexibility in designing school-based health education programs, highlighting the importance of tailoring educational content to the strengths of each method. Future studies should investigate whether there are even more effective strategies for enhancing this type of knowledge.

## Conclusion

Our study compared the most widely used educational approaches on this issue and demonstrated that direct instruction and workshop have similar effectiveness. However, direct instruction is more cost-efficient, given that it implies only a few and cheap materials, while the workshop and abbreviated workshop methods used here require more materials that are also more expensive. These findings have practical implications for health promotion in dengue, both in Argentina and elsewhere, given the choice of the educational method may depend more on contextual factors, such as the specific concepts being taught, or the available materials, rather than on a presumed superiority of one approach over the other.

## Supporting information

**S1 Fig. Linear mixed model assumption diagnostics.**
(TIF)

**S1 Table. Questionnaire items.** List of questions included in the questionnaire used in the study.
(XLSX)

**S2 Table. Sample size and percentage of cases shared across the three timepoints, per each timepoint.**
(XLSX)

**S3 Table. Linear mixed models between groups comparisons.**
(XLSX)

**S4 Table. Linear mixed models within groups comparisons.**
(XLSX)

**S5 Table. Linear mixed models between groups comparisons for significant questions.**
(XLSX)

**S6 Table. Linear mixed models within groups comparisons for significant questions.**
(XLSX)

## Acknowledgments

We would like to thank principals, teachers and families of the participating public schools: School 694 -Eduardo Bassi, School 837 – Daniel Eytan, School 200 – San Ignacio de Loyola, School 862 B° 2000 Has, School 875 -Mercedes G de Taratutty, School 947 B° 25 de Mayo, School 954 Altos del Paraná, School 164-Educar por la Paz, School 615-Mariano Moreno, School 711 B° Hermoso, School 778-Pablo Areguati. Also, a special thanks to the Puerto Iguazú Schools Supervisor Bs. Mónica Elorz. We would like to acknowledge the support of Mundo Sano´s team in the Puerto Iguazú Office, as well as Milagros Girart & Manuel Espinosa from the headquarters in Buenos Aires for their valuable input. AI was used for translation.

## Author contributions

**Conceptualization:** Julia Hermida, Carolina Goizueta, Maria Victoria Periago, Carolina Lopez Ferloni.

**Data curation:** Julia Hermida, Carolina Goizueta, Federico Giovannetti.

**Formal analysis:** Federico Giovannetti.

**Funding acquisition:** Maria Victoria Periago, Carolina Lopez Ferloni.

**Investigation:** Julia Hermida, Carolina Goizueta, Catalina Canosa, Carolina Lopez Ferloni.

**Methodology:** Julia Hermida, Maria Victoria Periago, Carolina Lopez Ferloni.

**Project administration:** Julia Hermida, Carolina Goizueta, Maria Victoria Periago, Carolina Lopez Ferloni.

**Resources:** Carolina Lopez Ferloni.

**Supervision:** Julia Hermida, Carolina Goizueta, Maria Victoria Periago, Carolina Lopez Ferloni.

**Visualization:** Federico Giovannetti.

**Writing – original draft:** Julia Hermida.

**Writing – review & editing:** Carolina Goizueta, Federico Giovannetti, Catalina Canosa, Maria Victoria Periago, Carolina Lopez Ferloni.

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
