## [Decision Letter · Decision Letter 0]

2 Sep 2025

Dear Dr. Hermida,

We look forward to receiving your revised manuscript.

Kind regards,

Rajib Chowdhury, M.Sc.; MPH

Academic Editor

PLOS ONE

Journal Requirements:

Additional Editor Comments:

Reviewer #1:

Reviewer #2:

Reviewers' comments:

Reviewer's Responses to Questions

**Comments to the Author**

1. Is the manuscript technically sound, and do the data support the conclusions?

Reviewer #1: Partly

Reviewer #2: Yes

2. Has the statistical analysis been performed appropriately and rigorously?

Reviewer #1: Yes

Reviewer #2: Yes

3. Have the authors made all data underlying the findings in their manuscript fully available?

Reviewer #1: Yes

Reviewer #2: Yes

4. Is the manuscript presented in an intelligible fashion and written in standard English?

Reviewer #1: Yes

Reviewer #2: Yes

Reviewer #1: Title: Suggest to revise the title to reflect the overall aim of the study.

Introduction: Study justifications and objectives are clear.

Methods:

- What is the age of the repondents? Please justify the decision to select this age group.

- Please specify what are the outcome measures studied. Why was it chosen? Any behaviour change outcome, apart from knowledge?

Results:

- Table 2: What mean score is presented here? What is the distribution of the score? State whether the scores were normal/not normally distributed.

- Table 3. Please state what is the reference group in the analysis.

Discussion and Conclussion:

- Will the results be applicable to other age groups?

- How will the findings inform the planning and implementation of dengue health education in Argentina?

Please revise these sentences to improve structure:

- Page 2; line 18

- Page 4; line 78, 79

Reviewer #2: This is a well written manuscript on dengue education strategies targeting children age 10 years of age. The authors compared talk vs abbreviated workshop vs workshop and found no significant difference. Rather education and retention of specific concepts may depend more on the context.

Major

The abstract mentioned that children learned 3 dengue concepts and retained 2 however this is not expanded upon in the full manuscript.

Line 49: Suggest amending the sentence. Reference 8 does not suggest that education is superior to legislation.

Line 53: Qualify that this is the risk in the local context as different countries have different population at risk

Minor

40-41 Arguably even if there is specific treatment for dengue, the most effective approach will still be prevention

41 Aedes aegypti is not the only vector for dengue

**Do you want your identity to be public for this peer review?** For information about this choice, including consent withdrawal, please see our Privacy Policy

Reviewer #1: No

Reviewer #2: No

---

## [Author Response · Author response to Decision Letter 1]

19 Sep 2025

Dear Editor and reviewers,

Thank you for the detailed lecture on our manuscript, which has contributed significantly to improving it. You will find our answers below each of your comments, in blue.

Comments to the Author

1. Is the manuscript technically sound, and do the data support the conclusions?

Reviewer #1: Partly

Reviewer #2: Yes

2. Has the statistical analysis been performed appropriately and rigorously?

Reviewer #1: Yes

Reviewer #2: Yes

3. Have the authors made all data underlying the findings in their manuscript fully available?

Reviewer #1: Yes

Reviewer #2: Yes

4. Is the manuscript presented in an intelligible fashion and written in standard English?

Reviewer #1: Yes

Reviewer #2: Yes

5. Review Comments to the Author

Reviewer #1: Title: Suggest to revise the title to reflect the overall aim of the study.

Following the reviewer's suggestion, we have changed the title “Direct instruction and workshops are equally effective in fostering children’s dengue knowledge” to: “Analyzing the comparative effectiveness of educational methods to foster children’s dengue knowledge.”

Introduction: Study justifications and objectives are clear.

Methods:

- What is the age of the repondents? Please justify the decision to select this age group.

Respondents are 10-year-olds. We included 10-year-old children because at this age they are literate, which allows both written knowledge assessments and workshops in which written resources can be used to work. In addition, the workshop material was pedagogically designed for 10-year-olds. All this information was added in the new version of the manuscript in the section “Participants”.

- Please specify what are the outcome measures studied. Why was it chosen? Any behaviour change outcome, apart from knowledge?

The outcome measure study is dengue knowledge. While recognizing the importance of behavior change for dengue prevention, we did not measure it because our study was aimed at comparing educational strategies in terms of their effectiveness to increase dengue knowledge (which is necessary, but not sufficient to produce behavioral change). On page 7, line 140 we added a sentence clarifying that “No behavioral change measure was included.”

Also, on page 14, lines, 293-295 the previous version clarified that further studies are needed to understand how knowledge translates into behavioral change. Now we moved that idea to lines 284-287, and included it as an essential limitation of our study that future studies should address.

Results:

- Table 2: What mean score is presented here? What is the distribution of the score?

The mean value presented in Table 2 refers to the mean of the total score grouped by time and experimental group. The “Measures” section as well as the table 2 note was expanded to clarify this.

State whether the scores were normal/not normally distributed.

The Score variable is a count measure constructed as the sum of all correct answers. Therefore, we did not expect raw values to necessarily follow a normal distribution, since count data are typically discrete, bounded at zero, and skewed. For this reason, we did not evaluate normality of the raw Score variable and, instead, went for a GLMM and tested normality of residuals among other diagnostic tests (GLMM does not assume normality of the dependent variable, but instead assumes normality of residuals). In order to provide more clarity, we added an explanation of model diagnostics in the “Analysis” section and added the GLMM residuals assumptions checks in the Supplementary data file.

- Table 3. Please state what is the reference group in the analysis.

In all models, the Talk group is the reference group. This was added in the table note.

Discussion and Conclussion:

- Will the results be applicable to other age groups?

This is a relevant question. Knowledge retention in the long term depends on long-term working memory systems. The neural basis of those systems relies on brain development, which occurs during the first two decades of life. Thus, the capacity to retain knowledge (i.e., to learn more dengue concepts) depends heavily on age during primary school. Therefore, we ignore whether our results will be valid for children at other developmental ages. Further studies are needed to check that. We have now stated that in the discussion section.

- How will the findings inform the planning and implementation of dengue health education in Argentina?

In order to answer this question, we have added lines 293-309 in the Discussion section.

Please revise these sentences to improve structure:

- Page 2; line 18

- Page 4; line 78, 79

Both sentences were corrected, thanks!

Reviewer #2: This is a well written manuscript on dengue education strategies targeting children age 10 years of age. The authors compared talk vs abbreviated workshop vs workshop and found no significant difference. Rather education and retention of specific concepts may depend more on the context.

Major

The abstract mentioned that children learned 3 dengue concepts and retained 2 however this is not expanded upon in the full manuscript.

Thanks for this comment. The data to conclude data is shown on Table 1. We have now expanded the explanation of this interpretation of the results in the “Discussion” section.

Line 49: Suggest amending the sentence. Reference 8 does not suggest that education is superior to legislation.

Thanks for this comment, the sentence was corrected.

Line 53: Qualify that this is the risk in the local context as different countries have different population at risk

Thanks for this comment. The sentence, as well as the reference, were deleted as they refer to Mexico but we do not have data on dengue prevalence by age groups in Argentina.

Minor

40-41 Arguably even if there is specific treatment for dengue, the most effective approach will still be prevention

41 Aedes aegypti is not the only vector for dengue

Right. To solve both minor comments, we have modified that sentence as follows: “The most effective approach to dengue is prevention, which involves controlling the virus's vector, that in the case of Argentina, is the Aedes aegypti mosquito [6].”

---

## [Decision Letter · Decision Letter 1]

26 Nov 2025

Dear Dr. Hermida,

Thank you for submitting your manuscript to PLOS ONE. After careful consideration, we feel that it has merit but does not fully meet PLOS ONE’s publication criteria as it currently stands. Therefore, we invite you to submit a revised version of the manuscript that addresses the points raised during the review process.

We look forward to receiving your revised manuscript.

Kind regards,

Rajib Chowdhury, M.Sc.; MPH

Academic Editor

PLOS ONE

Journal Requirements:

Reviewers' comments:

Reviewer's Responses to Questions

**Comments to the Author**

Reviewer #3: (No Response)

Reviewer #4: All comments have been addressed

Reviewer #5: All comments have been addressed

2. Is the manuscript technically sound, and do the data support the conclusions?

Reviewer #3: Yes

Reviewer #4: Yes

Reviewer #5: Yes

3. Has the statistical analysis been performed appropriately and rigorously?

Reviewer #3: Yes

Reviewer #4: Yes

Reviewer #5: Yes

4. Have the authors made all data underlying the findings in their manuscript fully available?

Reviewer #3: Yes

Reviewer #4: Yes

Reviewer #5: Yes

5. Is the manuscript presented in an intelligible fashion and written in standard English?

Reviewer #3: Yes

Reviewer #4: Yes

Reviewer #5: Yes

Reviewer #3: A well-structured manuscript written with limited data. The results are encouraging in a way that children at their growing age can acquire knowledge about dengue and may translate it into basis of their behavior change, though not sufficient. Education and knowledge about dengue in early growing age, indeed, have potential to contribute in prevention of dengue transmission in dengue endemic regions, no matter which part of the world.

I have minor suggestions to improvise the manuscript:

- Check and simplify the sentences in lines

o 81-85

- Use of the words “direct instruction”, “Lecture” and “talks” are interchangeably used in the manuscript. I feel this has to be clearly mentioned in the introduction or in the methodology and more precisely use a single term throughout the text.

- In line 153, “time x group” can be replaced with “time and group combined”

- In line 157, “ there was more than 80% overlap between the children who participated in each evaluation”, does this mean, there were children who did not take part in baseline survey but participated in T1 and T2 surveys? If yes, this need to be explicitly explained in the methodology.

- In lines 168 and 170, What is “H” stands for? If it is “Kruskal–Wallis test statistic”, then this has to be mentioned in Analysis subsection under Methodology that you did the test to assess among the groups. The same comments and suggestions for F-statistics, as the values are shown in line 186.

- In lines 174-175, you mentioned “Likewise, after one month, boys and girls reduce their knowledge but do so in the same way in all three groups (p < 0.001)”. You assessed the gender effect on your outcome, need to be mentioned in the methodology section.

- Do you think “Incidence Rate Ratio” would be better option to explain the comparison between the groups than the“β” co-efficient in GLMM?

Reviewer #4: (No Response)

Reviewer #5: Analyzing the comparative effectiveness of educational methods to foster children’s dengue knowledge

Well written/corrected manuscript. However, I have following suggestions.

1. Key words: Dengue is the topic but Dengue is not among the key words. If 6 words is limitation then remove Communicable Diseases and insert Dengue.

2. Why the n (sample size decreased abruptly in T2 in comparison to base line and T1?

3. Conclusion Line 308-309: However, few studies addressed the comparative effectiveness of educational interventions on dengue. Suggestion: Conclusion should include Authors findings only. Hence this line should be removed. But it might be placed in discussion section.

4. Line 309-311: Our study compared the most widely used educational approaches on this issue and demonstrated that direct instruction and workshop have similar effectiveness. Suggestion: If different approaches have similar effectiveness then please mention in a line, which method was cost effective, so that government/s may implement that in cost effective way.

**Do you want your identity to be public for this peer review?** For information about this choice, including consent withdrawal, please see our Privacy Policy

Reviewer #3: No

Reviewer #4: **Yes:** Ariful Basher

Reviewer #5: **Yes:** Murari Lal Das

---

## [Author Response · Author response to Decision Letter 2]

26 Dec 2025

Dear Editor and reviewers,

Thank you for the detailed lecture on our manuscript, which has contributed significantly to improving it. You will find our answers below each of your comments, in red.

Review Comments to the Author

Reviewer #3: A well-structured manuscript written with limited data. The results are encouraging in a way that children at their growing age can acquire knowledge about dengue and may translate it into basis of their behavior change, though not sufficient. Education and knowledge about dengue in early growing age, indeed, have potential to contribute in prevention of dengue transmission in dengue endemic regions, no matter which part of the world.

I have minor suggestions to improvise the manuscript:

- Check and simplify the sentences in lines 81-85

Thanks, sentences were corrected.

- Use of the words “direct instruction”, “Lecture” and “talks” are interchangeably used in the manuscript. I feel this has to be clearly mentioned in the introduction or in the methodology and more precisely use a single term throughout the text.

We agree. In this new version we used direct instruction across all the manuscript, only leaving “talk” in specific places, but not to refer to study groups.

- In line 153, “time x group” can be replaced with “time and group combined”

It was replaced

- In line 157, “ there was more than 80% overlap between the children who participated in each evaluation”, does this mean, there were children who did not take part in baseline survey but participated in T1 and T2 surveys? If yes, this need to be explicitly explained in the methodology.

Yes, we wanted to mean that. To clarify this point, we have added in the analysis section the following sentences:

“Moreover, given that GLMMs allow us to control unbalanced group sizes, there were children in T1 and T2 who did not take part in baseline, and vice versa, children in baseline that did not take part in T1 or T2. The percentage of children by group, at each time, that were also evaluated in the other two timepoints was calculated.”

Also, we have added in the Results section the following data: “(…) 17.34% of children in T1 and T2 not included in baseline”.

- In lines 168 and 170, What is “H” stands for? If it is “Kruskal–Wallis test statistic”, then this has to be mentioned in Analysis subsection under Methodology that you did the test to assess among the groups. The same comments and suggestions for F-statistics, as the values are shown in line 186.

To clarify this point, we have added the following sentences in the Analysis section:

“To assess the similarity of the different questionnaire versions in terms of scores and resolution times, we used a Kruskal-Wallis H test. The same test was implemented to compare the duration of the intervention activities and the number of participants included in them.”

And this at the end of the section we have added this sentence:

“The F-statistic was used for all GLMMs.”

- In lines 174-175, you mentioned “Likewise, after one month, boys and girls reduce their knowledge but do so in the same way in all three groups (p < 0.001)”. You assessed the gender effect on your outcome, need to be mentioned in the methodology section.

Thanks for alerting us about that mistake. We have not assessed the gender effect on our outcome as we wanted to check which intervention was most effective independently of gender. Thus, we have changed “boys and girls” by children in that sentence.

- Do you think “Incidence Rate Ratio” would be better option to explain the comparison between the groups than the“β” co-efficient in GLMM?

We thank the reviewer for the technical question. IRR as well as other exponential-based transformations could be a suitable option for informing the difference between the assessed groups. However, in the present study we decided to only inform the β for the sake of comparison with previous studies in similar populations in the country, in which the β estimate was informed.

Reviewer #4: (No Response)

Reviewer #5: Analyzing the comparative effectiveness of educational methods to foster children’s dengue knowledge

Well written/corrected manuscript. However, I have following suggestions.

1. Key words: Dengue is the topic but Dengue is not among the key words. If 6 words is limitation then remove Communicable Diseases and insert Dengue.

Thanks! Dengue was included as a keyword

2. Why the n (sample size decreased abruptly in T2 in comparison to base line and T1?

Baseline and T1 have slight variations in sample size due to the exclusion of unreliable questionnaires (protocols with scores below 5). Overall, sample size was maintained between Baseline and T1 because data was taken during the same day: as children were already in the school during the whole day, there was no attrition between those two measures. Instead, T2 was a measure taken one month later, in order to evaluate knowledge maintenance. At T2 various children were absent at schools during the days in which we collected the measures. Unfortunately, returning to schools some weeks later to complete data collection would have introduced other confusing variables given that knowledge retention is highly affected by time. Thus, in T2 we decided to keep only the data that was taken exactly one month later.

To clarify this point we added a note in Table 2.

3. Conclusion Line 308-309: However, few studies addressed the comparative effectiveness of educational interventions on dengue. Suggestion: Conclusion should include Authors findings only. Hence this line should be removed. But it might be placed in discussion section.

This sentence was removed

4. Line 309-311: Our study compared the most widely used educational approaches on this issue and demonstrated that direct instruction and workshop have similar effectiveness. Suggestion: If different approaches have similar effectiveness then please mention in a line, which method was cost effective, so that government/s may implement that in cost effective way.

Thanks for this important comment. We have added that line in the conclusion section.

---

## [Decision Letter · Decision Letter 2]

13 Jan 2026

Analyzing the comparative effectiveness of educational methods to foster children’s dengue knowledge

PONE-D-25-39030R2

Dear Dr. Hermida,

We’re pleased to inform you that your manuscript has been judged scientifically suitable for publication and will be formally accepted for publication once it meets all outstanding technical requirements.

Kind regards,

Rajib Chowdhury, M.Sc.; MPH

Academic Editor

PLOS One

Additional Editor Comments (optional):

Reviewers' comments:

Reviewer's Responses to Questions

**Comments to the Author**

Reviewer #3: All comments have been addressed

Reviewer #4: All comments have been addressed

2. Is the manuscript technically sound, and do the data support the conclusions?

Reviewer #3: Yes

Reviewer #4: Yes

3. Has the statistical analysis been performed appropriately and rigorously?

Reviewer #3: Yes

Reviewer #4: I Don't Know

4. Have the authors made all data underlying the findings in their manuscript fully available?

Reviewer #3: Yes

Reviewer #4: Yes

5. Is the manuscript presented in an intelligible fashion and written in standard English?

Reviewer #3: Yes

Reviewer #4: Yes

Reviewer #3: Only one comment:

In figures 1 and 2, the group name "talk" need to be changed to "direct instruction"

Reviewer #4: (No Response)

**Do you want your identity to be public for this peer review?** For information about this choice, including consent withdrawal, please see our Privacy Policy

Reviewer #3: No

Reviewer #4: **Yes:** Ariful Basher

---

## [Editor Report · Acceptance letter]

PONE-D-25-39030R2

PLOS One

Dear Dr. Hermida,

I'm pleased to inform you that your manuscript has been deemed suitable for publication in PLOS One. Congratulations! Your manuscript is now being handed over to our production team.

Kind regards,

on behalf of

Dr. Rajib Chowdhury

Academic Editor

PLOS One